# Fairness in Multi-Agent Sequential Decision-Making

**Chongjie Zhang and Julie A. Shah**
Computer Science and Artificial Intelligence Laboratory
Massachusetts Institute of Technology
Cambridge, MA 02139
{chongjie,julie_a_shah}@csail.mit.edu

## Abstract

We define a fairness solution criterion for multi-agent decision-making problems, where agents have local interests. This new criterion aims to maximize the worst performance of agents with a consideration on the overall performance. We develop a simple linear programming approach and a more scalable game-theoretic approach for computing an optimal fairness policy. This game-theoretic approach formulates this fairness optimization as a two-player zero-sum game and employs an iterative algorithm for finding a Nash equilibrium, corresponding to an optimal fairness policy. We scale up this approach by exploiting problem structure and value function approximation. Our experiments on resource allocation problems show that this fairness criterion provides a more favorable solution than the utilitarian criterion, and that our game-theoretic approach is significantly faster than linear programming.

## Introduction

Factored multi-agent MDPs [4] offer a powerful mathematical framework for studying multi-agent sequential decision problems in the presence of uncertainty. Its compact representation allows us to model large multi-agent planning problems and to develop efficient methods for solving them. Existing approaches to solving factored multi-agent MDPs [4] have focused on the utilitarian solution criterion, i.e., maximizing the sum of individual utilities. The computed utilitarian solution is optimal from the perspective of the system where the performance is additive. However, as the utilitarian solution often discriminates against some agents, it is not desirable for many practical applications where agents have their own interests and fairness is expected. For example, in manufacturing plants, resources need to be fairly and dynamically allocated to work stations on assembly lines in order to maximize the throughput; in telecommunication systems, wireless bandwidth needs to be fairly allocated to avoid "unhappy" customers; in transportation systems, traffic lights are controlled so that traffic flow is balanced.

In this paper, we define a fairness solution criterion, called *regularized maximin fairness*, for multi-agent MDPs. This criterion aims to maximize the worst performance of agents with a consideration on the overall performance. We show that its optimal solution is Pareto-efficient. In this paper, we will focus on centralized joint policies, which are sensible for many practical resource allocation problems. We develop a simple linear programming approach and a more scalable game-theoretic approach for computing an optimal fairness policy. This game-theoretic approach formulates this fairness optimization for factored multi-agent MDPs as a two-player, zero-sum game. Inspired by theoretical results that two-player games tend to have a Nash equilibrium (NE) with a small support [7], we develop an iterative algorithm that incrementally solves this game by starting with a small subgame. This game-theoretic approach can scale up to large problems by relaxing the termination condition, exploiting problem structure in factored multi-agent MDPs, and applying value function approximation. Our experiments on a factory resource allocation problem show that this

fairness criterion provides a more favorable solution than the utilitarian criterion [4], and our game-theoretic approach is significantly faster than linear programming.

## Multi-agent decision-making model and its fairness solution

We are interested in multi-agent sequential decision-making problems, where agents have their own interests. We assume that agents are cooperating. Cooperation can be proactive, e.g., sharing resources with other agents to sustain cooperation that benefits all agents, or passive, where agents' actions are controlled by a thirty party, as with centralized resource allocation. We use a factored multi-agent Markov decision processes (MDP) to model multi-agent sequential decision-making problems [4]. A factored multi-agent MDP is defined by a tuple $\langle I, \mathbf{X}, \mathbf{A}, T, \{R_i\}_{i \in I}, b \rangle$, where

$I = \{1, \ldots, n\}$ is a set of agent indices.

$\mathbf{X}$ is a state space represented by a set of state variables $\mathbf{X} = \{X_1, \ldots, X_m\}$. A state is defined by a vector $\mathbf{x}$ of value assignments to each state variable. We assume the domain of each variable is finite.

$\mathbf{A} = \times_{i \in I} A_i$ is a finite set of joint actions, where $A_i$ is a finite set of actions available for agent $i$. The joint action $\mathbf{a} = \langle a_1, \ldots, a_n \rangle$ is defined by a vector of individual action choices.

$T$ is the transition model. $T(\mathbf{x}'|\mathbf{x}, \mathbf{a})$ specifies the probability of transitioning to the next state $\mathbf{x}'$ after a joint action $\mathbf{a}$ is taken in the current state $\mathbf{x}$. As in [4], we assume that the transition model can be factored and compactly represented by a dynamic Bayesian network (DBN).

$R_i(\mathbf{x}_i, \mathbf{a}_i)$ is a local reward function of agent $i$, which is defined on a small set of variables $\mathbf{x}_i \subseteq \mathbf{X}$ and $\mathbf{a}_i \subseteq \mathbf{A}$.

$b$ is the initial distribution of states.

This model allows us to exploit problem structures to represent exponentially-large multi-agent MDPs compactly. Unlike factored MDPs defined in [4], which have one single reward function represented by a sum of partial reward functions, this multi-agent model has a local reward function for each agent. From the multi-agent perspective, existing approaches to factored MDPs [4] essentially aim to compute a control policy that maximizes the utilitarian criterion (i.e., the sum of individual utilities). As the utilitarian criterion often provides a solution that is not fair or satisfactory for some agents (e.g., as shown in the experiment section), it may not be desirable for problems where agents have local interests.

In contrast to the utilitarian criterion, an egalitarian criterion, called *maximin fairness*, has been studied in networking [1, 9], where resources are allocated to optimize the worst performance. This egalitarian criterion exploits the maximin principle in Rawlsian theory of justice [14], maximizing the benefits of the least-advantaged members of society. In the following, we will define a fairness solution criterion for multi-agent MDPs by adapting and combining the maximin fairness criterion and the utilitarian criterion. Under this new criterion, an optimal policy for multi-agent MDPs aims to maximize the worst performance of agents with a consideration on the overall performance.

A joint stochastic policy $\pi : \mathbf{X} \times \mathbf{A} \to \Re$ is a function that returns the probability of taking joint action $a \in \mathbf{A}$ for any given state $x \in \mathbf{X}$. The utility of agent $i$ under a joint policy $\pi$ is defined as its infinite-horizon, total discounted reward, which is denoted by

$$\psi(i, \pi) = \mathbf{E}[\sum_{t=0}^{\infty} \lambda^t R_i(\mathbf{x}^t, \mathbf{a}^t)|\pi, b]. \tag{1}$$

where $\lambda$ is the discount factor, the expectation operator $\mathbf{E}(\cdot)$ averages over stochastic action selection and state transition, $b$ is the initial state distribution, and $\mathbf{x}^t$ and $\mathbf{a}^t$ are the state and the joint action taken at time $t$, respectively.

To achieve both fairness and efficiency, our goal for a given multi-agent MDP is to find a joint control policy $\pi^*$, called a **regularized maximin fairness policy**, that maximizes the following objective value function

$$V(\pi) = \min_{i \in I} \psi(i, \pi) + \frac{\epsilon}{n} \sum_{i \in I} \psi(i, \pi), \tag{2}$$

where $n = |I|$ is the number of agents and $\epsilon$ is a strictly positive real number, chosen to be arbitrary small. [1] This fairness objective function can be seen as a lexicographic aggregation of the egalitarian criterion (min) and utilitarian criterion (sum of utilities) with priority to egalitarianism. This fairness criterion can be also seen as a particular instance of the weighted Tchebycheff distance with respect to a reference point, a classical secularization function used to generate compromise solutions in multi-objective optimization [16]. Note that the optimal policy under the egalitarian (or maximin) criterion alone may not be Pareto efficient, but the optimal policy under this regularized fairness criterion is guaranteed to be Pareto efficient.

**Definition 1.** *A joint control policy $\pi$ is said to be* **Pareto efficient** *if and only if there does not exist another joint policy $\pi'$ such that the utility is at least as high for all agents and strictly higher for at least one agent, that is, $\nexists \pi', \forall i, \psi(i, \pi') \geq \psi(i, \pi) \wedge \exists i, \psi(i, \pi') > \psi(i, \pi)$.*

**Proposition 1.** *A regularized maximin fairness policy $\pi^*$ is Pareto efficient.*

*Proof.* We can prove by contradiction. Assume regularized maximin fairness policy $\pi^*$ is not Pareto efficient. Then there must exist a policy $\pi$ such that $\forall i, \psi(i, \pi) \geq \psi(i, \pi^*) \wedge \exists i, \psi(i, \pi) > \psi(i, \pi^*)$. Then $V^{\pi} = \min_{i \in I} \psi(i, \pi) + \frac{\epsilon}{n} \sum_{i \in I} \psi(i, \pi) > \min_{i \in I} \psi(i, \pi^*) + \frac{\epsilon}{n} \sum_{i \in I} \psi(i, \pi^*) = V^{\pi^*}$, which contradicts the pre-condition that $\pi^*$ is a regularized maximin fairness policy. $\square$

In this paper, we will mainly focus on centralized policies for multi-agent MDPs. This focus is sensible because we assume that, although agents have local interests, they are also willing to cooperate. Many practical problems modeled by multi-agent MDPs use centralized policies to achieve fairness, e.g., network bandwidth allocation by telecommunication companies, traffic congestion control, public service allocation, and, more generally, fair resource allocation under uncertainty. On the other hand, we can derive decentralized policies for individual agents from a maximin fairness policy $\pi^*$ by marginalizing it over the actions of all other agents. If the maximin fairness policy is deterministic, then the derived decentralized policy profile is also optimal under the regularized maximin fairness criterion. Although such a guarantee generally does not hold for stochastic policies, as indicated by the following proposition, the derived decentralized policy is a bounded solution in the space of decentralized policies under the regularized maximin fairness criterion.

**Proposition 2.** *Let $\pi^{c^*}$ be an optimal centralized policy and $\pi^{dec^*}$ be an optimal decentralized policy profile under the regularized maximin fairness criterion. Let $\pi^{dec}$ be an decentralized policy profile derived from $\pi^{c^*}$ by marginalization. The values of policy $\pi^{c^*}$ and $\pi^{dec}$ provides bounds for the value of $\pi^{dec^*}$, that is,*

$$V(\pi^{c^*}) \geq V(\pi^{dec^*}) \geq V(\pi^{dec}).$$

The proof of this proposition is quite straightforward. The first inequality holds because any decentralized policy profile can be converted to a centralized policy by product, and the second inequality holds because $\pi^{dec^*}$ is an optimal decentralized policy profile. When bounds provided by $V(\pi^{c^*})$ and $V(\pi^{dec})$ are close, we can conclude that $\pi^{dec}$ is almost an optimal decentralized policy profile under the regularized maximin fairness criterion.

In this paper, we are primarily concerned with total discounted rewards for an infinite horizon, but the definition, analysis, and computation of regularized maximin fairness can be adapted to a finite horizon with an undiscounted sum of rewards. In the next section, we will present approaches to computing the regularized maximin fairness policy for infinite-horizon multi-agent MDPs.

## Computing Regularized Maximin Fairness Policies

In this section, we present two approaches to computing regularized maximin fairness policies for multi-agent MDPs: a simple linear programming approach and a game theoretic approach. The former approach is adapted from the linear programming formulation of single-agent MDPs. The latter approach formulates this fairness problem as a two-player zero-sum game and employs an iterative search method for finding a Nash equilibrium that contains a regularized maximin fairness policy. This iterative algorithm allows us to scale up to large problems by exploiting structures in multi-agent MDPs and value function approximation and employing a relaxed termination condition.

**A linear programming approach**

For a multi-agent MDP, given a joint policy and the initial state distribution, frequencies of visiting state-action pairs are uniquely determined. We use $f_\pi(\mathbf{x}, \mathbf{a})$ to denote the total discounted probability, under the policy $\pi$ and initial state distribution $b$, that the system occupies state $\mathbf{x}$ and chooses action $\mathbf{a}$. Using this frequency function, we can rewrite the expected total discount rewards as follows, using $f_\pi(\mathbf{x}, \mathbf{a})$:

$$\psi(i, \pi) = \sum_{\mathbf{x}} \sum_{\mathbf{a}} f_\pi(\mathbf{x}, \mathbf{a}) R_i(\mathbf{x}_i, \mathbf{a}_i), \tag{3}$$

where $\mathbf{x}_i \subseteq \mathbf{x}$ and $\mathbf{a}_i \subseteq \mathbf{a}$.

Since the dynamics of a multi-agent MDPs is Markovian, as it is for the single-agent MDP, we can adapt the linear programming formulation of single-agent MDPs for finding an optimal centralized policy for multi-agent MDPs under the regularized maximin fairness criterion as follows:

$$\max_{f} \quad \min_{i \in I} \sum_{\mathbf{x}} \sum_{\mathbf{a}} f(\mathbf{x}, \mathbf{a}) R_i(\mathbf{x}_i, \mathbf{a}_i) + \frac{\epsilon}{n} \sum_{i \in I} \sum_{\mathbf{x}} \sum_{\mathbf{a}} f(\mathbf{x}, \mathbf{a}) R_i(\mathbf{x}_i, \mathbf{a}_i)$$

$$\text{s.t.} \quad \sum_{\mathbf{a}} f(\mathbf{x}', \mathbf{a}) = b(\mathbf{x}') + \sum_{\mathbf{x}} \sum_{\mathbf{a}} \lambda T(\mathbf{x}'|\mathbf{x}, \mathbf{a}) f(\mathbf{x}, \mathbf{a}), \qquad \forall \mathbf{x}' \in \mathbf{X}$$

$$f(\mathbf{x}, \mathbf{a}) \geq 0, \qquad \text{for all } \mathbf{a} \in \mathbf{A} \text{ and } \mathbf{x} \in \mathbf{X}. \tag{4}$$

Constraints are included to ensure that $f(\mathbf{x}, \mathbf{a})$ is well-defined. The first set of constraints require that the probability of visiting state $\mathbf{x}'$ is equal to the initial probability of state $\mathbf{x}'$ plus the sum of all probabilities of entering into state $s'$. We linearize this program by introducing another variable $z$, which represents the minimum expected total discounted reward among all agents, as follows:

$$\max_{f} \quad z + \frac{\epsilon}{n} \sum_{i \in I} \sum_{\mathbf{x}} \sum_{\mathbf{a}} f(\mathbf{x}, \mathbf{a}) R_i(\mathbf{x}_i, \mathbf{a}_i)$$

$$\text{s.t.} \quad z \leq \sum_{\mathbf{x}} \sum_{\mathbf{a}} f(\mathbf{x}, \mathbf{a}) R_i(\mathbf{x}_i, \mathbf{a}_i), \qquad \forall i \in I$$

$$\sum_{\mathbf{a}} f(\mathbf{x}', \mathbf{a}) = b(\mathbf{x}') + \sum_{\mathbf{x}} \sum_{\mathbf{a}} \lambda T(\mathbf{x}'|\mathbf{x}, \mathbf{a}) f(\mathbf{x}, \mathbf{a}), \qquad \forall \mathbf{x}' \in \mathbf{X}$$

$$f(\mathbf{x}, \mathbf{a}) \geq 0, \qquad \text{for all } \mathbf{a} \in \mathbf{A} \text{ and } \mathbf{x} \in \mathbf{X}. \tag{5}$$

We can employ existing linear programming solvers (e.g., the simplex method) to compute an optimal solution $f^*$ for problem (5) and derive a policy $\pi^*$ from $f^*$ by normalization:

$$\pi(\mathbf{x}, \mathbf{a}) = \frac{f(\mathbf{x}, \mathbf{a})}{\sum_{\mathbf{a} \in \mathbf{A}} f(\mathbf{x}, \mathbf{a})}. \tag{6}$$

Using Theorem 6.9.1 in [13], we can easily show that the derived policy $\pi^*$ is optimal under the regularized maximin fairness criterion. This linear programming approach is simple, but is not scalable for multi-agent MDPs with large state spaces or large numbers of agents. This is because the number of constraints of the linear program is $|\mathbf{X}| + |I|$. In the next sections, we present a more scalable game-theoretic approach for large multi-agent MDPs.

**A game-theoretic approach**

Since the fairness objective function in (2) can be turned to a maximin function, inspired by von Neumann's minimax theorem, we can formulate this optimization problem as a two-player zero-sum game. Motivated by theoretical results that two-player games tend to have a Nash equilibrium (NE) with a small support, we develop an iterative algorithm for solving zero-sum games.

Let $\Pi^S$ and $\Pi^D$ be the set of stochastic Markovian policies and deterministic Markovian policies, respectively. As shown in [13], every stochastic policy can be represented by a convex combination of deterministic policies and every convex combination of deterministic policies corresponds to a stochastic policy. Specifically, for any stochastic policy $\pi^s \in \Pi^s$, we can represent $\pi^s = \sum_i p_i \pi_i^d$ using some set of $\{\pi_1^d, \ldots, \pi_k^d\} \subset \Pi^D$ with probability distribution $p$.

**Algorithm 1:** An iterative approach to computing the regularized maximin fairness policy

**1** Initialize a zero-sum game $G(\bar{\Pi}^D, \bar{I})$ with small subsets $\bar{\Pi}_s^D \subset \Pi^D$ and $\bar{I} \subset I$ ;
**2 repeat**
**3**     $(p^*, q^*, V^*) \leftarrow$ compute a Nash equilibrium of game $G(\bar{\Pi}^D, \bar{I})$ ;
**4**     $(\pi^d, V_p) \leftarrow$ compute the best-response deterministic policy against $q^*$ in $G(\Pi^D, I)$ ;
**5**     **if** $V_p > V^*$ **then** $\bar{\Pi}^D \leftarrow \bar{\Pi}^D \cup \{\pi^d\}$ ;
**6**     $(i, V_q) \leftarrow$ compute the best response against $p^*$ among all agents $I$;
**7**     **if** $V_q < V^*$ **then** $\bar{I} \leftarrow \bar{I} \cup \{i\}$ ;
**8**     **if** $G(\bar{\Pi}^D, \bar{I})$ *changes* **then** expand its payoff matrix with $U(\pi^d, i)$ for new pairs $(\pi^d, i)$ ;
**9 until** *game* $G(\bar{\Pi}^D, \bar{I})$ *converges*;
**10 return** the regularized maximin fairness policy $\pi_{p^*}^s = p^* \cdot \bar{\Pi}^D$ ;

Let $U(\pi, i) = \psi(i, \pi) + \frac{\epsilon}{n} \sum_{j \in I} \psi(j, \pi)$. We can construct a two-player zero-sum game $G(\Pi^D, I)$ as follows: the maximizing player, who aims to maximize the value of the game, chooses a deterministic policy $\pi^d$ from $\Pi^D$; the minimizing player, who aims to minimizing the value of the game, chooses an agent indexed by $i$ in multi-agent MDPs from $I$; and the payoff matrix has an entry $U(\pi^d, i)$ for each pair $\pi^d \in \Pi^D$ and $i \in I$. The following proposition shows that we can compute the regularized minimax fairness policy by solving $G(\Pi^D, I)$.

**Proposition 3.** *Let the strategy profile* $(p^*, q^*)$ *be a NE of the game* $G(\Pi^D, I)$ *and the stochastic policy* $\pi_{p^*}^s$ *which is derived from* $(p^*, q^*)$ *with* $\pi_{p^*}^s(\mathbf{x}, \mathbf{a}) = \sum_i p_i^* \pi_i^d(\mathbf{x}, \mathbf{a})$, *where* $p_i^*$ *is the ith component of* $p^*$, *i.e., the probability of choosing the deterministic policy* $\pi_i^d \in \Pi^D$. *Then* $\pi_{p^*}^s$ *is a regularized maximin fairness policy,*

*Proof.* According to von Neumann's minimax theorem, $p^*$ is also the maximin strategy for the zero-sum game $G(\Pi^D, I)$.

$$
\begin{aligned}
\min_i U(\pi_{p^*}^s, i) &= \min_i \sum_j p_j^* U(\pi_j^d, i) \quad \text{(let } \pi_{p^*}^s = \sum_j p_j^* \pi_j^d \text{)} \\
&= \min_q \sum_j \sum_i p_j^* q_i U(\pi_j^d, i) \quad \text{(there always exists a pure best response strategy)} \\
&= \max_p \min_q \sum_j \sum_i p_j q_i U(\pi_j^d, i) \quad (p^* \text{ is the maximin strategy)} \\
&\geq \max_p \min_i \sum_j p_j U(\pi_j^d, i) \quad \text{(consider } i \text{ as a pure strategy)} \\
&= \max_{\pi_p} \min_i U(\pi_p, i) \quad \text{(let } \pi_p = \sum_j p_j \pi_j^d \text{)}
\end{aligned}
$$

By definition, $\pi_{p^*}^s$ is a regularized maximin fairness policy. $\qquad \square$

As the game $G(\Pi^D, I)$ is usually extremely large and computing the payoff matrix of the game $G(\Pi^D, I)$ is also non-trivial, it is impossible to directly use linear programming to solve this game. On the other hand, existing work, such as [7] that analyzes the theoretical properties of the NE of games drawn from a particular distribution, shows that support sizes of Nash equilibria tend to be balanced and small, especially for $n = 2$. Prior work [11] demonstrated that it is beneficial to exploit these results in finding a NE, especially in 2-player games. Inspired by these results, we develop an iterative method to compute a fairness policy, as shown in Algorithm 1.

Intuitively, Algorithm 1 works as follows. It starts by computing a NE for a small subgame (Line 3) and then checks whether this NE is also a NE of the whole game (Line 4-7); if not, it expands the subgame and repeats this process until a NE is found for the whole game.

Line 1 initializes a small sub game of the original game, which can be arbitrary. In our experiments, it is initialized with a random agent and a policy maximizing this agent's utility. Line 3 solves the two-player zero-sum game using linear programming or any other suitable technique. $V^*$ is the maximin

value of this subgame. The best response problem in Line 4 is to find a deterministic policy $\pi$ that maximizes the following payoff:

$$U(\pi, q^*) = \sum_{i \in \bar{I}} q_i^* U(\pi, i) = \sum_{i \in \bar{I}} q_i^* [\psi(i, \pi) + \frac{\epsilon}{n} \sum_{j \in I} \psi(j, \pi)] = \sum_{i \in I} (q_i^* + \frac{\epsilon}{n}) \psi(i, \pi)$$

Solving this optimization problem is equivalent to finding the optimal policy of a regular MDP with a reward function $R(\mathbf{x}, \mathbf{a}) = \sum_{i \in I} (q_i^* + \frac{\epsilon}{n}) R_i(\mathbf{x}_i, \mathbf{a}_i)$. We can use the dual linear programming approach [13] for this MDP, which outputs the visitation frequency function $f_{\pi^d}(\mathbf{x}, \mathbf{a})$ representing the optimal policy. This representation facilitates the computation of the payoff $U(\pi_i^d, i)$ using Equation 3. $V_p = \sum_i q_i^* U(\pi^d, i)$ is the maximizing player's utility of its best response against $q^*$.

Line 5 checks if the best response $\pi^d$ is strictly better than $p^*$. If this is true, we can infer that $p^*$ is not the best response against $q^*$ in the whole game and $\pi^d$ must not be in $\bar{\Pi}^D$, which is then added to $\bar{\Pi}^D$ to expand the subgame.

Line 6 finds the minimizing player's best response against $p^*$, which minimizes the payoff of the maximizing player. Note that there always exists a pure best response strategy. So we formulate this best response problem as follows:

$$\min_{i \in I} U(\pi_{p^*}, q) = \min_{i \in I} \sum_j p_j^* U(\pi_j^d, i), \tag{7}$$

where $\pi_{p^*}$ is the stochastic policy corresponding to probability distribution $p^*$. We can solve this problem by directly searching for the agent $i$ that yields the minimum utility with linear time complexity. Similar to Line 5, Line 7 checks if the minimizing player strictly preferred $i$ to $q^*$ against $p^*$ and expands the subgame if needed. This algorithm terminates when the subgame does not change.

**Proposition 4.** *Algorithm 1 converges to a regularized maximin fairness policy.*

*Proof.* The convergence of this algorithm follows immediately because there exists a finite number of deterministic Markovian policies and agents for a given multi-agent MDP. The algorithm terminates if and only if neither of the **If** conditions of Line 5 and 7 hold. This situation indicates no player strictly prefers a strategy out of the support of its current strategy, which implies $(p^*, q^*)$ is a NE of the whole game $G(\bar{\Pi}^D, \bar{I})$. Using Proposition 3, we conclude that Algorithm 1 returns a regularized maximin fairness policy. $\square$

Algorithm 1 shares some similarities with the double oracle algorithm proposed in [8] for iteratively solving zero-sum games. The double oracle method is motivated by Benders decomposition technique, while our iterative algorithm exploits properties of Nash equilibrium, which leads to a more efficient implementation. For example, unlike our algorithm, the double oracle method checks if the computed best response MDP policy exists in the current sub-game by comparison, which is time-consuming for MDP policies with a large state space.

**Scaling the game-theoretic approach**

Both linear programming and the game-theoretic approach suffer scalability issues for large problems. In multi-agent MDPs, the state space is exponential with the number of state variables and the action space is exponential with the number of agents. This results in an exponential number of variables and constraints in linear program formulation. In this section, we will investigate methods to scale up the game-theoretic approach.

The major bottleneck of the iterative algorithm is the computation of the best response policy (Line 4 in Algorithm 1). As discussed in the previous section, this optimization is equivalent to finding the optimal policy of a regular MDP with reward function $R(\mathbf{x}, \mathbf{a}) = \sum_i (q_i^* + \frac{\epsilon}{n}) R_i(\mathbf{x}_i, \mathbf{a}_i)$. Due to the exponential state-action space, exact algorithms (e.g., linear programming) are impractical in most cases. Fortunately, this MDP is essentially a factored MDP [4] with a weighted sum of partial reward functions. We can use existing approximate algorithms [4] to solve factored MDPs, which exploit both factored structures in the problem and value function approximation. For example, the approximate linear programming approach for factored MDPs can provide efficient policies with up to an exponential reduction in computation time.

| #C | #R | #N | Time-LP | Time-GT | Sol-LP | Sol-GT |
|----|-----|------|---------|---------|--------|--------|
| 4  | 12  | 7E4  | 68.22s  | 11.43s  | 157.67 | 154.24 |
| 4  | 20  | 3E5  | 22.39m  | 35.27s  | 250.59 | 239.87 |
| 5  | 10  | 4E5  | 89.77m  | 48.56s  | 104.33 | 97.48  |
| 5  | 20  | 6E6  | -       | 4.98m   | -      | 189.62 |
| 6  | 18  | 5E7  | -       | 43.36m  | -      | 153.63 |

| C   | MPE    | Utilitarian | Fairness |
|-----|--------|-------------|----------|
| 1   | 180.41 | 117.44      | 250.59   |
| 2   | 198.45 | 184.20      | 250.59   |
| 3   | 216.49 | 290.69      | 250.59   |
| 4   | 234.53 | 444.08      | 250.59   |
| Min | **108.22** | **68.32** | **157.67** |

Table 1: Performance in sample problems with different cell sizes and total resoureces

Table 2: A comparison of three criteria in a 4-agent 20-resource problem

A few subtleties are worth noting when approximate linear programming is employed. First, the best response's utility $V_p$ should be computed by evaluating the computed approximate policy against $q^*$, instead of directly using the value from the approximate value function. Otherwise, the convergence of Algorithm 1 will not be guaranteed. Similarly, the payoff $U(\pi^d, i)$ should be calculated through policy evaluation. Second, existing approximate algorithms for factored MDPs usually output a deterministic policy $\pi^d(\mathbf{x})$ that is not represented by the visitation frequency function $f_\pi(\mathbf{x}, \mathbf{a})$. In order to facilitate the policy evaluation, we may convert a policy $\pi^d(\mathbf{x})$ to a frequency function $f_{\pi^d}(\mathbf{x}, \mathbf{a})$. Note that $f_{\pi^d}(\mathbf{x}, \mathbf{a}) = 0$ for all $\mathbf{a} \neq \pi^d(\mathbf{x})$. For other state-action pairs, we can compute their visitation frequencies by solving the following equation:

$$f_{\pi^d}(\mathbf{x}', \pi^d(\mathbf{x}')) = b(\mathbf{x}') + \sum_{\mathbf{x}} T(\mathbf{x}'|\mathbf{x}, \mathbf{a}) f_{\pi^d}(\mathbf{x}, \pi^d(\mathbf{x})). \tag{8}$$

This equation can be approximately but more efficiently solved using an iterative method, similar to the MDP value iteration. Finally, Algorithm 1 is still guaranteed to converge, but may return a sub-optimal solution.

We can also speed up Algorithm 1 by relaxing its termination condition, which essentially reduces the number of iterations. We can use the termination condition $V_p - V_q < \epsilon$, which turns the iterative approach into an approximation algorithm.

**Proposition 5.** *The iterative approach using the termination condition $V_p - V_q < \epsilon$ has bounded error $\epsilon$.*

*Proof.* Let $V^{opt}$ be the value of the regularized maximin fairness policy and $V(\pi^*)$ be the value of the computed policy $\pi^*$. By definition, $V^{opt} \geq V(\pi^*)$. Following von Neumann's minimax theorem, we have $V_p \geq V^{opt} \geq V_q$. Since $V_q$ is the value of the minimizing player's best response against $\pi^*$, $V^{opt} \geq V(\pi^*) \geq V_q \geq V_p + \epsilon \geq V^{opt} + \epsilon$. □

## Experiments

One motivated domain for our work is resource allocation in a pulse-line manufacturing plant. In a pulse-line factory, the manufacturing process of complex products is divided into several stages, each of which contains a set of tasks to be done in a corresponding work cell. The overall performance of a pulse line is mainly determined by the worse performance of work cells. Considering dynamics and uncertainty of the manufacturing environment, we need to dynamically allocate resources to balance the progress of work cells in order to optimize the throughput of the pulse line.

We evaluate our fairness solution criterion and its computation approaches, linear programming (LP) and the game-theoretic (GT) approach with approximation, on this resource allocation problem. For simplicity, we focus on managing one type of resource. We view each work cell in a pulse line as an agent. Each agent's state is represented by two variables: task level (i.e., high or low) and the number of local resources. An agent's next task level is affected by the current task levels of itself and the previous agent. An action is defined on a directed link between two agents, representing the transfer of one-unit resource from one agent to another. There is another action for all agents: "no change". We assume only neighboring agents can transfer resources. An agent's reward is measured by the number of partially-finished products that will be processed during two decision points, given its current task level and resources. We use a discount factor $\lambda = 0.95$. We use the approximate linear programming technique presented in [4] for solving factored MDPs generated in the GT approach. We used Java for our implementation and Gurobi 2.6 [5] for solving linear programming and ran experiments on a 2.4GHz Intel Core i5 with 8Gb RAM.

Table 1 shows the performance of linear programming and the game-theoretic approach in different problems by varying the number of work cells #C and total resources #R. The third column #N = $|\mathbf{X}||\mathbf{A}|$ is the state-action space size. We can observe that the game-theoretic approach is significantly faster than linear programming. This speed improvement is largely due to the integration of approximate linear programming, which exploits the problem structure and value function approximation. In addition, the game-theoretic approach is scalable well to large problems. With 6 cells and 18 resources, the size of the state-action space is around $5 \cdot 10^7$. The last two columns show the minimum expected reward among agents, which determines the performance of the pulse line. The game-theoretic approach only has a less than 8% loss over the optimal solution computed by LP.

We also compare the regularized maximin fairness criterion against the utilitarian criterion (i.e., maximizing the sum of individual utility) and Markov perfect equilibrium (MPE). MPE is an extension of Nash equilibrium to stochastic games. One obvious MPE in our resource allocation problem is that no agent transfers its resources to other agents. We evaluated them in different problems, but the results are qualitatively similar. Table 2 shows the performance of all work cells under the optimal policy of different criteria in a problem with 4 agents and 20 resources. The fairness policy balanced the performance of all agents and provided a better solution (i.e., a greater minimum utility) than other criteria. The perfection of the balance is due to the stochasticity of the computed policy. Even in terms of the sum of utilities, the fairness policy has only a less than 4% loss over the optimal policy under the utilitarian criterion. The utilitarian criterion generated a highly skewed solution with the lowest minimum utility among the three criteria. In addition, we can observe that, under the fairness criterion, all agents performed better than those under MPE, which suggests that cooperation is beneficial for all of them in this problem.

## Related Work

When using centralized policies, our multi-agent MDPs can be also viewed as multi-objective MDPs [15]. Recently, Ogryczak et al. [10] defined a compromise solution for multi-objective MDPs using the Tchebycheff scalarization function. They developed a linear programming approach for finding such compromise solutions; however, this is computationally impractical for most real-world problems. In contrast, we develop a more scalable game-theoretic approach for finding fairness solutions by exploiting structure in multi-agent factored MDPs and value function approximation.

The notion of maximin fairness is also widely used in the field of networking, such as bandwidth sharing, congestion control, routing, load-balancing and network design [1, 9]. In contrast to our work, maximin fairness in networking is defined without regularization, only addresses one-shot resource allocation, and does not consider the dynamics and uncertainty of the environment.

Fair division is an active research area in economics, especially social choice theory. It is concerned with the division of a set of goods among several people, such that each person receives his or her due share. In the last few years, fair division has attracted the attention of AI researchers [2, 12], who envision the application of fair division in multi-agent systems, especially for multi-agent resource allocation [3, 6]. Fair division theory focuses on proportional fairness and envy-freeness. Most existing work in fair division involves a static setting, where all relevant information is known upfront and is fixed. Only a few approaches deal with dynamics of agent arrival and departures [6, 17]. In contrast to our model and approach, these dynamic approaches to fair division do not address uncertainty, or other dynamics such as changes of resource availability and users' resource demands.

## Conclusion

In this paper, we defined a fairness solution criterion, called *regularized maximin fairness*, for multi-agent decision-making under uncertainty. This solution criterion aims to maximize the worse performance among agents while considering the overall performance of the system. It is finding applications in various domains, including resource sharing, public service allocation, load balance, and congestion control. We also developed a simple linear programming approach and a more scalable game-theoretic approach for computing the optimal policy under this new criterion. This game-theoretic approach can scale up to large problems by exploiting the problem structure and value function approximation.

## Footnotes

[1] In some applications, we may choose proper large $\epsilon$ to trade off fairness and the overall performance.

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
