[Reviews · NeurIPS 2014]

Submitted by Assigned_Reviewer_8

The paper proposes a fairer optimization criterion, “regularized maximin”, for centralized multi-agent MDPs. The idea, taken from the networking literature is elegant. The authors also propose an iterative optimization method that scales somewhat better than linear programming.

Specific comments:
1. The description of the transition model, lines 69-79, seems unnecessarily detailed. Do the theorems depend on the details of the DBN used to model transitions? Does the proposed algorithm?

2. The authors state that most of the speedup from LP to GT is due to using Guestrin's approximate LP. Given this, it would be better to compare GT against both LP and Approximate LP — to better understand the usefulness of the proposed game-theoretic algorithm.

Criticisms:
1. I’m not sure NIPS is the best venue for this paper, as opposed to AAAI or AAMAS. There is no learning.

2. The paper seems incremental. The contributions are: (i) regularizing the previously introduced maximin criterion and (ii) the game-theoretic algorithm — who’s usefulness is obscured by the lack of a competitive baseline.
Summary: Overall this is a nice, well-written paper. Strikes me as too incremental.

Submitted by Assigned_Reviewer_19

This paper proposes a policy selection criterion (regularized maximin fairness) for multi-agent factored MDPs that is equivalent to the most utilitarian of egalitarian policies. The paper then presents an algorithm to find such a policy based on framing the problem as a zero sum problem between a utilitarian player and an egalitarian player. The algorithm operates over multiple iterations of solving standard utilitarian multi-agent factored MDPs.

The paper is well written with nice theoretical proofs, and the idea of viewing the problem as a zero sum game is compelling. However, I believe this view has unnecessarily complicated the solution. Also, I am worried that the problem itself is not strongly motivated.

- Since the problem being solved is ultimately centralized, the need for a different selection criteria must be due to a need for greater model expressiveness or conciseness. Its not clear to me that the problems couldn’t just be modeled in a different way (perhaps with additional states) using a standard global reward function that expresses the goal they wish to maximize (i.e. throughput). No argument is made that these problems don’t arise only due to lack of modeling skill. Even more importantly, utility theory (as used by MDPs) relies on the expected utility hypothesis (risk neutral), but it seems to me that preferences over policies given by regularized maximin fairness breaks this assumption by preferring (motivationally) to be risk adverse.

For example, suppose we have a problem where two agents can choose either to A) receive (1, 1) or B) flip a fair coin and randomly receive (0, 4) or (4, 0). Each individual’s utility preference is to flip the coin, but the reward received by centralized maximin would have a value of 0 if the coin was chosen. Or course, this is not how the maximin criteria is defined, but is how it should be defined given the motivation. All of the examples given (assembly plan, telecommunications, traffic lights) would prefer the (1,1) solution, hence my concern with the motivation, and if the regularized maximin fairness is the proper means of modeling these problems. Even if the problem is repeated (so expectation makes more sense) agents would still want to be risk adverse.

- I believe the algorithm should be viewed as column generation to solve the zero-sum game. The variables are weights on deterministic joint-policies (i.e. p* along with some q* for which the policy maximizes) such that the resulting solution is a convex combination of these policies. The objective becomes the value of the minimum player. Column generation proceeds as per the best response problem in Line 4. Where q* = the convex combination of q*’s that make up the current basis. One immediate benefit of this view, is that line 3 needs not be solved each time, instead the new column can be added to the current basis and one (or more) of the old columns can be removed.

- I don’t understand why line 4 of the algorithm 1 is considered a single-agent MDP. Doesn’t the agent need to optimize over all agent’s actions, which is only efficient because this is a factored multi-agent MDP? Calling this a single-agent MDP is misleading.

- Pet peeve: Use of the word “obviously” to avoid explanation. This is basically argument by ridicule (“if you don’t understand this step, then you’re stupid”) and should be avoided. If you don’t want to explain a step, don’t explain it. No need to insult your reader.

- The discussion on decentralized policies and Proposition 2 seems out of place in this paper. DecPOMDPs are quite a different beast, and its not clear how this work would help decentralized policies. Proposition 2 is a general statement for any optimal centralized policy and is already known.
Summary: The paper is well written with nice theoretical proofs, and the idea of viewing the problem as a zero sum game is compelling. However, I believe this view has unnecessarily complicated the solution and I am worried that the problem itself is not strongly motivated.

Submitted by Assigned_Reviewer_22

PAPER SUMMARY

This paper poses an interesting problem, which frequently arises in many real-world situations where utility/resource needs to be fairly allocated to the participating agents: Instead of strictly maximizing the sum of individual utilities, this paper aims to maximize the worse performance of an individual with a consideration on the overall performance, thus ensuring stability.

In particular, the multi-agent environment is modeled as a QMDP and the main contributions of this work are highlighted below:

1. The authors introduce the regularized maximin fairness criterion, which computationalizes the above conceptual aim. The optimal decision-theoretic policy (i.e., regularized maximin fairness policy) that maximizes this criterion is theoretically demonstrated to be Pareto efficient & solvable using linear programming.

2. Interestingly, the authors further establish a dual view for the regularized fairness maximin policy which corresponds to Nash equilibrium (NE) in a constructed two-player zero-sum stochastic game. This implies we can solve for the regularized maximin policy by solving for the corresponding NE, which is more scalable.

3. By taking advantage of the previously established result that NE strategy profile of the game tends to have small support with high probability, the authors propose a clever approach of iteratively searching for the NE in a bottom-up manner starting from some small sub-games & checking if the sub-game’s NE is also the NE of the whole game (which happens with high probability).

4. The devised algorithm is then rigorously shown to converge to a regularized maximum fairness policy and can be modified to compromise with an arbitrary loss bound epsilon.

SIGNIFICANCE & ORIGINALITY

This paper is novel in both its problem & the proposed solution. Remarkably, the proposed solution is both scalable (as carefully detailed in the experiments) and rigorously analyzed. I consider this a significant contribution.

QUALITY

Both the theory & experiments are carefully demonstrated and of high quality.

CLARITY

The paper is very well-written.
Summary: An interesting paper with nice theoretical results which are insightful in both theory & practice.
Author Feedback
Author rebuttal: Reviewer_19:

Thank you for your thoughtful comments. We would like to clarify your concerns. Firstly, our definition of the maximin fairness is well motivated by practical problems. This is because most of these problems are multi-stage and repeated and, as you mentioned, utility theory relies on the expected utility and agents are risk neutral. Therefore, in your example, when the problem is repeated, it is more sensible to choose flipping the fair coin, because it has higher expected utility. Our fairness definition is consistent with this choice.

We did realize that there was an alternative way to define fairness, which exchanges the expectation operator and the minimum operator in our fairness definition (i.e., Equation 2). With this new definition, (1, 1) will be preferred in your example. The main problem of this definition is that it is numerically difficult to compute the optimal solution. In fact, by exploiting Jensen’s inequality, we can show the optimal solution of the maximin fairness provides a bounded solution for this alternative fairness definition. We will add discussions to the final version of the paper about this alternative definition.

Secondly, our iterative algorithm is actually more efficient than the single column generation approach you described, especially when there are more agents. This is because our iterative algorithm applies column generation to both players in zero-sum games, while the approach you described applies the column generation technique on just one player. Of course, when the number of agents is small, it may not be necessary to apply column generation in computing q* (the optimal mixture of reward function). In fact, the major computational complexity of each iteration for both algorithm is in solving an MDP given a mixed reward function to compute the best response policy \pi_{p*} (Line 4), while the computation cost in Line 3 is relatively trivial.

Reviewer_22:

Thank you for your positive comments.

Reviewer_8:

We strongly disagree with the reviewer’s criticisms. To our best knowledge, this paper is the first work that studies fairness in multi-agent planning under uncertainty. Although sharing some similar “notion” of maximin fairness in networking, our work is totally different from existing work on fairness in networking in all aspects, including problem, definition, and algorithms. Therefore, this paper is not incremental to existing work in fairness.

The reviewer also seemed to misunderstand the GT algorithm. The GT algorithm allows us to exploit existing approximate LP techniques in factored MDPs so that we can compute fairness policies for large problems. It is not clear yet if we can use approximate LP to directly compute regularized maximin fairness policies. Therefore, in this paper, we compared the GT algorithm with approximate LP to LP and showed that it could scale to large problems.

Finally, according to the NIPS CFP, we believe that this paper is highly relevant to NIPS, because it is in the NIPS technical areas of MDPs, planning, and multi-agent coordination.